# Adjusted Comparison of Outcomes between Patients from CARTITUDE-1 versus Multiple Myeloma Patients with Prior Exposure to PI, Imid and Anti-CD-38 from a German Registry

**DOI:** 10.3390/cancers13235996

**Published:** 2021-11-29

**Authors:** Maximilian Merz, Hartmut Goldschmidt, Parameswaran Hari, Mounzer Agha, Joris Diels, Francesca Ghilotti, Nolen J. Perualila, Jedelyn Cabrieto, Benjamin Haefliger, Henrik Sliwka, Jordan M. Schecter, Carolyn C. Jackson, Yunsi Olyslager, Muhammad Akram, Tonia Nesheiwat, Lenka Kellermann, Sundar Jagannath

**Affiliations:** 1Cell Therapy and Hemostaseology, Department of Hematology, University Hospital of Leipzig, 04103 Leipzig, Germany; 2Internal Medicine V and National Center for Tumor Diseases, University Clinic Heidelberg, 69120 Heidelberg, Germany; hartmut.goldschmidt@med.uni-heidelberg.de; 3Medical College of Wisconsin, Milwaukee, WI 53226, USA; phari@mcw.edu; 4School of Medicine, University of Pittsburgh, Pittsburgh, PA 15213, USA; aghame@upmc.edu; 5Janssen Pharmaceutica NV, 2340 Beerse, Belgium; JDIELS@its.jnj.com (J.D.); NPeruali@its.jnj.com (N.J.P.); JCabriet@ITS.JNJ.com (J.C.); YOLYSLA@its.jnj.com (Y.O.); 6Janssen-Cilag SpA, 20093 Cologno Monzese, Italy; FGhilott@its.jnj.com; 7Cilag GmbH International, 6300 Zug, Switzerland; BHaeflig@its.jnj.com; 8Janssen-Cilag Pharma GmbH, 1020 Vienna, Austria; HSliwka@its.jnj.com; 9Janssen R&D, Raritan, NJ 08869, USA; jschecte@ITS.JNJ.com (J.M.S.); CJacks66@its.jnj.com (C.C.J.); 10Legend Biotech USA, Inc., Piscataway, NJ 08854, USA; Muhammad.Akram@legendbiotech.com (M.A.); tonia.nesheiwat@legendbiotech.com (T.N.); 11Oncology Information Service (O.I.S), 79098 Freiburg, Germany; kellermann@OncologyInformationService.com; 12Mount Sinai Medical Center, New York, NY 10029, USA; Sundar.Jagannath@mountsinai.org

**Keywords:** relapsed or refractory multiple myeloma, ciltacabtagene autoleucel, cilta-cel, CARTITUDE-1, chimeric antigen receptor T-cell therapy, CAR-T, indirect treatment comparison, adjusted comparison

## Abstract

**Simple Summary:**

There is an urgent need to develop new treatments for patients with relapsed/refractory multiple myeloma (RRMM) to address unmet medical needs. Chimeric antigen receptor T-cell (CAR-T) therapy is a novel approach with the potential for long-term disease control. Ciltacabtagene autoleucel (cilta-cel) is a CAR-T treatment studied in patients with RRMM in the CARTITUDE-1 clinical trial and has shown clinically important effects. However, CARTITUDE-1 was a single arm study. The current study compared outcomes for cilta-cel with an external cohort of German patients that are similar to the ones in CARTITUDE-1 to compare the effectiveness of cilta-cel versus established clinical practice. To overcome potential bias, individual patient data were used to adjust for the differences in patient characteristics between cohorts. The results showed substantially better outcomes for cilta-cel on both overall survival and the time to next treatment. These findings highlight cilta-cel’s potential as a novel, effective treatment to address unmet treatment needs.

**Abstract:**

Ciltacabtagene autoleucel (cilta-cel) is a Chimeric antigen receptor T-cell therapy with the potential for long-term disease control in heavily pre-treated patients with relapsed/refractory multiple myeloma (RRMM). As cilta-cel was assessed in the single-arm CARTITUDE-1 clinical trial, we used an external cohort of patients from the Therapie Monitor registry fulfilling the CARTITUDE-1 inclusion criteria to evaluate the effectiveness of cilta-cel for overall survival (OS) and time to next treatment (TTNT) vs. real-world clinical practice. Individual patient data allowed us to adjust the comparisons between both cohorts, using the inverse probability of treatment weighting (IPW; average treatment effect in the treated population (ATT) and overlap population (ATO) weights) and multivariable Cox proportional hazards regression. Outcomes were compared in intention-to-treat (HR, IPW-ATT: TTNT: 0.13 (95% CI: 0.07, 0.24); OS: 0.14 (95% CI: 0.07, 0.25); IPW-ATO: TTNT: 0.24 (95% CI: 0.12, 0.49); OS: 0.26 (95% CI: 0.13, 0.54)) and modified intention-to-treat (HR, IPW-ATT: TTNT: 0.24 (95% CI: 0.09, 0.67); OS: 0.26 (95% CI: 0.08, 0.84); IPW-ATO: TTNT: 0.26 (95% CI: 0.11, 0.59); OS: 0.31 (95% CI: 0.12, 0.79)) populations. All the comparisons were statistically significant in favor of cilta-cel. These results highlight cilta-cel’s potential as a novel, effective treatment to address unmet needs in patients with RRMM.

## 1. Introduction

Multiple myeloma (MM) is a hematological cancer that is characterized by the clonal proliferation of malignant plasma cells and the overproduction of M protein, an abnormal immunoglobulin [1]. MM is a highly heterogeneous cancer with a variable clinical course and substantial clinical burden that becomes progressively more severe [2,3,4]. MM accounts for 1% of all cancers worldwide and approximately 10% of hematological neoplasms [1]. In the European Union (EU) and the United States (US), approximately 50,000 patients are diagnosed with MM and 30,000 patients die due to MM each year [5].

Recent therapies, including immunomodulatory agents (IMiDs; thalidomide, lenalidomide and pomalidomide), proteasome inhibitors (PIs; bortezomib, ixazomib and carfilzomib) and more recently, monoclonal antibodies (daratumumab, isatuximab and elotuzumab) have substantially improved treatment outcomes for patients with MM in the last decade [6,7,8,9,10,11]. Despite these recent advancements in treatment, MM remains an incurable malignancy and most patients experience relapse and require additional therapy [2,3,4,5,6,7,8,9,10,11,12]. However, for patients that were previously exposed to PIs, IMiDs and anti-CD-38-antibodies, no standard of care exists and treatment outcomes with individualized therapies are poor [4,5,6,7,8,9,10,11,12,13]. New, more efficacious treatment options are required for these patients to extend survival, prevent disease progression and improve quality of life, addressing unmet medical needs [13,14,15].

In recent years, chimeric antigen receptor T-cell (CAR-T) therapy has emerged as a novel approach to treatment that may offer long-term control of the disease in certain hematological cancers [16]. Ciltacabtagene autoleucel (cilta-cel; JNJ–68284528) is an experimental form of CAR-T therapy that targets the B-cell maturation antigen (BCMA) [17], and which has been tested in CARTITUDE-1 (NCT03548207) [18]. CARTITUDE-1 is an open-label, single arm clinical trial that studies the safety and efficacy of cilta-cel for the treatment of relapsed/refractory MM (RRMM) in patients who have been administered at least three prior lines of therapy (LOTs) or who are double-refractory to an IMiD and a PI, and in prior LOTs, an IMiD, a PI and an anti-CD38 antibody (triple-class exposed) [18,19].

As the design of CARTITUDE-1 did not include a comparator group, adjusted comparisons of trial outcomes relative to an external cohort of similar patients allows for the estimation of clinical benefits relative to therapies used to treat patients in clinical practice, creating a hypothetical head-to-head trial. Therapie Monitor, run by Oncology Information Service, is a patient registry in Germany that contains a long-term longitudinal follow-up of patients diagnosed with multiple myeloma from diagnosis. The information available in Therapie Monitor allows us to identify a patient cohort fulfilling the CARTITUDE-1 inclusion criteria and hence can serve as an external control to CARTITUDE-1. Given the differences in treatment practices across countries and regions, the data from Therapie Monitor can help to contextualize findings from CARTITUDE-1 in a European setting. In this study, we conducted adjusted comparisons using individual patient data from CARTITUDE-1 and Therapie Monitor to compare the effectiveness of cilta-cel versus the currently available therapies in real-world clinical practice (RWCP) in triple-class exposed patients.

## 2. Materials and Methods

### 2.1. Data Sources

#### 2.1.1. Patients Treated with Ciltacabtagene Autoleucel in CARTITUDE-1

CARTITUDE-1 is an open-label, single arm phase 1b/2 clinical trial conducted to characterize the safety and efficacy of cilta-cel in adult patients with triple-class exposed RRMM. The study enrolled 113 patients from multiple centers in the US between July 2018 and October 2019. Enrolled patients received leukapheresis and the collected T-cells were used to generate the cilta-cel. Ninety-seven patients ultimately received cilta-cel; the remaining 16 patients were discontinued from the study due to death (9), withdrawal (5) and progressive disease (2). Further CARTITUDE-1 study details and outcomes have been previously published [18]. This study used updated data from CARTITUDE-1 as of February 2021 with a median follow-up of 18 months [20].

#### 2.1.2. Patients Receiving Treatments from Real World Clinical Practice in Therapie Monitor

Therapie Monitor is a real-world database initiated in January 2016 and maintained to date by Oncology Information Service (OIs). The database contains fully anonymized data on approximately 4000 patients with RRMM from a representative sample of 108 German centers. Data before 2016 are collected retrospectively from initial diagnosis with longitudinal follow-ups ongoing [21]. The documentation of electronic case report forms is performed retrospectively in Therapie Monitor. Records include diagnostic and treatment details captured during routine clinical care, either as structured data (e.g., the International Classification of Diseases ICD-9 and ICD-10 codes, laboratory values, medication orders, medication administrations) or unstructured data (e.g., abstracted data obtained from patient files documents including confirmed diagnosis, fluorescence in situ hybridization (FISH) cytogenetics and karyotyping at diagnosis, and stem cell transplant). The data source allowed us to derive outcome data on overall survival (OS) and time to next treatment (TTNT). As an assessment of response and progression are not recorded, TTNT was used as a proxy for progression-free survival (PFS). For the present study, patients who fulfilled the key eligibility criteria as in CARTITUDE-1 were selected, i.e., if they (1) had been exposed to a PI, an IMiD and an anti-CD38 antibody as part of previous therapy (either from different monotherapies or combination regimens); (2) had received at least 3 prior lines of MM treatment regimens (RRMM as defined by IMWG consensus criteria); (3) received a subsequent therapy after becoming triple class exposed; and (4) had an ECOG score < 2. Note: CARTITUDE-1 eligibility criteria allowed the inclusion of tri-exposed patients with <3 prior lines of therapy (when patients were double refractory to an IMiD and a PI). However, as all enrolled patients in CARTITUDE-1 had received at least 3 prior lines of therapy, only similar patients were included from Therapie Monitor, maximizing the comparability of the two cohorts. Additional CARTITUDE-1 eligibility criteria, e.g., absence of cardiac conditions, absence of prior history of central nervous system involvement or signs of meningeal involvement of multiple myeloma, could not be applied to the OIs cohort, as these are outside the scope of Therapie Monitor in the required high granularity. For the current study, follow-up until 31 December 2020 was used.

### 2.2. Analysis Populations and Design

Two analysis populations were defined, the intention-to-treat (ITT) and modified intention-to-treat (mITT) populations. For CARTITUDE-1, the ITT population corresponds to all patients enrolled in the trial, which was all patients that underwent apheresis (*N* = 113), while the mITT population corresponds to all patients that received cilta-cel (*N* = 97).

As data in Therapie Monitor were collected retrospectively, patients could have initiated more than one treatment line after meeting the eligibility criteria. In CARTITUDE-1, however, patients might have received cilta-cel after receiving other LOTs after fulfilling the eligibility criteria. Systematically including the first (or the last) available treatment line from the external cohort, after meeting the eligibility criteria, would induce selection bias, as the line of therapy is associated with the outcomes. For example, the last treatment line of patients in the registry for whom death was observed, generally have worse survival outcomes, as this line is, by definition, the last line available and hence, the approach is biased [22]. To overcome this, all treatment lines initiated after a patient met eligibility criteria were used for analysis in this study [23,24], as long as patients fulfilled the inclusion criteria at the start of the respective LOT. The unit of observation for the external cohort was the line of therapy within patients. Clustering of observations within patients was taken into account using the robust sandwich variance estimator [25]. The data included for the ITT population for Therapie Monitor was, therefore, composed of all LOTs meeting the eligibility criteria (*N* = 312). The mITT population was composed of all LOTs meeting the eligibility criteria, after excluding LOTs from patients with an event or who were censored within 52 days after treatment initiation (the average duration between apheresis and infusion in CARTITUDE-1, *N* = 223), in order to avoid survivor bias of patients not reaching infusion in CARTITUDE-1.

The index date, T_0_, was defined as the date when inclusion criteria were met. In CARTITUDE-1, this was the date of apheresis for the ITT population and the date of infusion for the mITT population. Index dates for patients in the Therapie Monitor registry was the date of treatment initiation for the ITT population, and the date of treatment initiation plus 52 days for mITT (see Appendix A).

### 2.3. Baseline Characteristics for Population Alignment

Comparisons of non-randomized populations can be biased due to imbalances in prognostic baseline characteristics if they are not adjusted for in the analyses. Potentially prognostic baseline characteristics were identified a priori by literature reviews and consulting with clinical experts. These were then rank ordered according to expected importance by clinical experts. The following factors, available in both data sources, were included in the analyses: refractory status, revised ISS stage (R-ISS) at index date, time to progression on last prior line, number of prior lines of treatment, ECOG status, age at index date, sex, average duration of prior lines and years since diagnosis. R-ISS stage was derived based on available values in cytogenetic risk (when tested), serum albumin and β2-microglobulin for both CARTITUDE-1 and Therapie Monitor (see Appendix A). The prognostic strength of these factors was explored using univariate and multivariable regression and imbalances were assessed using standardized mean differences (SMD), where values > 0.2 were considered indicative of potentially important differences [26]. As total plasmacytomas was not available in Therapie Monitor, it was not included in the analyses, despite being considered an important variable. Similarly, while comorbidities of patients are collected on a more general level in Therapie Monitor, these were not considered in the analysis.

### 2.4. Endpoints

The following two outcome measures were available for comparisons between cilta-cel and RWCP: TTNT and OS. Outcome definitions were aligned between CARTITUDE-1 and Therapie Monitor. TTNT was defined as the time from the index date to the initiation of the next therapy line or death, whichever occurred first. Patients who were still alive and did not initiate a next therapy line at the time of the data-cut, were censored at the last date they were known to be alive.

OS was measured as the time from the index date to the date of the patient’s death. If a patient was either alive or of unknown status, then data were censored at the date he or she was last known to be alive.

### 2.5. Statistical Methods

Individual patient data available for both CARTITUDE-1 and Therapie Monitor were pooled to conduct the analyses. Analyses of TTNT and OS were performed for ITT and mITT populations, and both unadjusted and adjusted comparisons are presented.

To address differences in baseline covariates between patient cohorts, inverse probability weighting (IPW) analyses were conducted. First, the propensity score for each subject was estimated using a multivariable logistic regression model. In a second step, different sets of weights were derived and were used in the weighted analyses. IPW was the primary approach employed in the analysis, more specifically patients were re-weighted using weights to derive the average treatment effect in the treated population (IPW-ATT) and the average treatment effect in the overlap population (IPW-ATO). The ATT approach kept CARTITUDE-1 cohort as observed, i.e., assigned them a weight of 1, and reweighted the Therapie Monitor cohort to make it similar to the trial population by assigning it a weight of *p*/(1 − *p*), where *p* is the propensity score predicting inclusion in the CARTITUDE-1 cohort. This gave patients in the Therapie Monitor cohort a higher weight if they are similar to CARTITUDE-1 patients and a lower weight if they were different. However, in case of limited overlap between patient populations, propensity scores and derived weights may become extreme, potentially resulting in biased and/or unstable estimates. The IPW-ATO can account for any limitations in population overlap. It can provide estimates for the target population with the most overlap in observed characteristics between treatments, by down-weighting observations in the tails of the propensity score distributions where there is insufficient overlap between both treatment cohorts [27,28].

Effectiveness of cilta-cel versus RWCP was evaluated using IPW weights and Multivariable Cox proportional hazards (PH) regression models. The cumulative effects of incrementally including covariates into the model one at a time were assessed through an inspection of changes in the adjusted comparison estimates between cilta-cel and RWCP, and the effects of the covariates in the fully adjusted models were reviewed. The variance was estimated using a robust sandwich variance estimator to account for the clustering of multiple treatment lines within patients. To estimate hazard ratios (HR) and its respective 95% CI for time-to-event outcomes using an IPW approach, a weighted Cox proportional hazards model was used.

The appropriateness of the proportional hazard assumption for time-to-event outcomes was assessed based on a visual inspection of the log-cumulative hazard plot and the Schoenfeld residuals plot, and performance of the Grambsch–Therneau test [29] (with a *p*-value less than 0.05 considered to indicate a violation of the assumption). A visual assessment was also conducted to carefully assess the shape of the curves over time.

Analyses were performed with SAS 9.4 (SAS Institute, Cary, NC, USA) and R version 4.0.3 (R Foundation for Statistical Computing, Vienna, Austria).

### 2.6. Role of the Sponsor

The sponsor implemented the design of the comparative study, the data analysis and interpretation and the writing of the manuscript. CARTITUDE-1 was funded and conducted by the sponsor.

## 3. Results

### 3.1. Patient Population

Two patient populations were analyzed, the ITT and mITT cohort. The ITT cohort consisted of 113 patients from CARTITUDE-1 and 312 treatment lines from 222 patients for Therapie Monitor (Figure 1). The mITT cohort for CARTITUDE-1 included 97 patients [18] and consisted of 223 treatment lines from 174 patients for Therapie Monitor (Figure 1).

Following the application of IPW-ATT weights to re-weight the Therapie Monitor population, the degree of differences between the cilta-cel and RWCP groups was reduced, though some imbalances still remained (details provided in Appendix A). The remaining, reduced imbalances after ATT weighting may still bias comparative results in both directions (in favor of cilta-cel for the R-ISS stage, ECOG status and age; against cilta-cel for refractory status, sex and average duration of prior lines). The application of IPW-ATO weights provided perfect balance between the groups (Appendix A).

### 3.2. Treatment Regimens Received in the Real-World Clinical Practice Cohort

The most commonly received treatment regimens in the RWCP cohort consisted of ixazomib–lenalidomide–dexamethasone (18%), pomalidomide–dexamethasone (15%), melphalan–prednisone (11%), Elotuzumab–lenalidomide–dexamethasone (8%) and bortezomib–dexamethasone (7%). Details of the 33 unique regimens used are provided in Appendix A.

### 3.3. Findings, ITT Analyses

#### 3.3.1. Comparison Results, Overall Survival

The unweighted median OS in the cilta-cel group had not been reached after 18 months of median follow-up, while the median OS in the RWCP group was 9.89 months (95% CI: 7.43, 12.81). Figure 2 (Panel a) summarizes the results from the unadjusted comparisons and the adjusted comparisons. The unadjusted HR comparing cilta-cel vs. RWCP was 0.29 (95% CI: 0.19, 0.43) in favor of cilta-cel. After the IPW-ATT-based adjustment, the HR comparing cilta-cel and RWCP was 0.14 (95% CI: 0.07, 0.25), while the IPW-ATO re-weighting showed an HR of 0.26 (95% CI: 0.13, 0.54). Based on the multivariable Cox proportional hazards regression models, the estimated HR was 0.29 (95% CI: 0.15, 0.58). All the estimates were significantly in favor of cilta-cel.

#### 3.3.2. Comparison Results, Time to Next Treatment

The unweighted median TTNT in the cilta-cel group had not been reached after 18 months of median follow-ups, while the median TTNT in the RWCP group was 6.21 months (95% CI: 5.09, 6.80). Figure 2 (Panel b) summarizes the results from the unadjusted comparisons and the adjusted comparisons. The unadjusted HR comparing cilta-cel and RWCP was 0.21 (95% CI: 0.15, 0.30) in favor of cilta-cel. Following IPW-ATT reweighting, the adjusted HR comparing cilta-cel and RWCP was 0.13 (95% CI: 0.07, 0.24). Using IPW-ATO re-weighting, the adjusted HR was 0.24 (95% CI: 0.12, 0.49); the HR applying multivariable Cox proportional hazards regression models was 0.20 (95% CI: 0.11, 0.39). All the estimates significantly favored cilta-cel.

### 3.4. Findings, mITT Analyses

Figure 3 presents a summary of the patient characteristics from the mITT populations. Compared to the RWCP cohort, the cilta-cel-treated patients were more likely to be quadruple- or penta-refractory (definitions provided in Appendix A), have an R-ISS stage of I or II, have progressed on a prior treatment regimen in less than 4 months, have received five or more prior LOTs, have an ECOG performance status of 0 (rather than 1), be <65 years of age, be six or more years past initial diagnosis and have an average duration of prior treatment lines of >17.61 months.

#### 3.4.1. Comparison Results, Overall Survival

The unweighted median OS in the cilta-cel group had not been reached after 18 months of median follow-up, while the median OS in the RWCP group was 11.86 months (95% CI: 9.00, NE) (Figure 4, Panel a). Figure 2 (Panel a) summarizes the results from the unadjusted comparisons and the adjusted comparisons. The unadjusted HR comparing cilta-cel and RWCP was 0.25 (95% CI: 0.16, 0.40) in favor of cilta-cel. When applying IPW-ATT re-weighting, the adjusted HR comparing cilta-cel and RWCP was 0.26 (95% CI: 0.08, 0.84). Following the IPW-ATO re-weighting, the adjusted HR comparing cilta-cel and RWCP was 0.31 (95% CI: 0.12, 0.79). When applying a multivariable Cox proportional hazards regression model, the HR was 0.16 (0.06, 0.42). All the estimates were significantly in favor of cilta-cel.

Figure 3 presents a summary of the covariate effects on OS derived from multivariable Cox PH regression analysis. While several factors were associated with statistically significant effects in univariate modeling, after adjusting for all the factors, those that remained significant were the R-ISS stage, the number of prior treatment lines and the ECOG status. While there was a statistically significant association observed for refractory status and the average duration of prior treatment lines in univariate modeling, there was only a non-significant association in the full model, indicating the collinearity of these characteristics with other factors.

#### 3.4.2. Comparison Results, Time to Next Treatment

The unweighted median TTNT in the cilta-cel group was not reached after 18 months of median follow-ups, while the median TTNT in the RWCP group was 6.54 months (95% CI: 4.76, 8.18) (Figure 4, Panel b). Figure 2 (Panel a) summarizes the results from the unadjusted comparisons and the adjusted comparisons. The unadjusted HR comparing cilta-cel and RWCP was 0.17 (95% CI: 0.11, 0.26). Following the IPW-ATT reweighting, the adjusted HR comparing cilta-cel and RWCP was 0.24 (95% CI: 0.09, 0.67). Following the IPW-ATO re-weighting, the adjusted HR was 0.26 (95% CI: 0.11, 0.59). Applying a multivariable Cox proportional hazards regression model produced an HR of 0.15 (0.07, 0.33) (Appendix A). All the estimates were significantly in favor of cilta-cel. All the findings for TTNT were consistent with the results for OS.

## 4. Discussion

While there have been improvements in recent years in terms of the treatment options for patients with multiple myeloma, there exists a pressing need for the development of novel therapies to reduce the unmet treatment needs of patients with triple-class exposed relapsed and refractory disease [13,14,15,16,17,18,19,20,21,22,23,24,25,26,27,28,29,30]. Cilta-cel has demonstrated both a manageable safety profile as well as durable treatment responses in CARTITUDE-1 in this patient population. However, given that no current standard of care nor equipoise in the selection of a comparator exists, CARTITUDE-1 was performed as a single arm trial. Hence, there emerges a need to compare the benefits of such a treatment relative to current clinical practice. In cases where head-to-head clinical trial data are unavailable, external control groups from real world data sources can serve as an extremely valuable source of information for clinicians as well as decision makers as a point of reference against which new therapies can be compared. Methods for adjusted comparisons are employed to overcome the lack of randomization and the associated possibility of confounding bias that might be present due to differences in baseline characteristics that are associated with patients’ outcomes.

In the current study, patient outcomes with cilta-cel, as observed in CARTITUDE-1, were compared with outcomes from similar patients treated with RWCP, as observed in Therapie Monitor in Germany. Therapie Monitor provides a representative sample of patients from German routine clinical practice and has excellent variable coverage, ensuring reliable comparisons [21,22,23,24,25,26,27,28,29,30,31,32]. This is further illustrated by the observed median OS and TTNT in Therapie Monitor of 9.89 and 6.21 months, respectively, consistent with outcomes in other sources and geographies [13,14,15,16,17,18,19,20,21,22,23,24,25,26,27,28,29,30].

After accounting for differences in the baseline characteristics between treatment groups using IPW and regression approaches, the estimates of effect for both OS and TTNT suggested considerable benefits associated with cilta-cel relative to the therapies used in real-world practice. The hazard ratios from these analyses show a reduced risk of death between 71 and 86% (depending on statistical approach) and an improvement in the time to next treatment of 76–87% (depending on the statistical approach), both representing clinically and statistically important effects of relevance to patients. The findings from this study are consistent both when looking at different analysis methods and patient populations, as well as when comparing with similar comparisons of cilta-cel vs. other external cohorts [33,34,35], showing the robustness and reliability of the findings, independent of the clinical setting and local real-world clinical practice.

The results of the current study should be considered in light of certain inherent limitations. First, as in any observational study, confounding for unobserved patient characteristics cannot be excluded. However, the range of covariates accounted for was broad, and included key clinical measures in the form of refractory status, R-ISS stage, time to disease progression on prior treatment lines, number of prior treatment lines, ECOG performance status, patient age and sex, average length of prior treatment lines and years passed since initial diagnosis. These helped to address the differences between groups in regard to factors associated with higher risk, some of which were more prevalent within the CARTITUDE-1 population (e.g., refractory disease and shorter time to disease progression on prior treatment lines) and some of which were more common in the Therapie Monitor cohort (e.g., older age and higher R-ISS). Second, while the analyses adjusted for a range of critical prognostic factors, limited confounding remained in the case of the ATT-IPW adjustment, indicating remaining bias in this analysis. It should also be noted that certain risk factors could not be included in the model (namely, extramedullary plasmacytomas, detailed comorbidities or cytogenetic risk) because this information was not routinely available in the patient’s charts. For example, in case of cytogenetic risk, its testing typically requires a bone marrow aspiration and is a costly procedure and, therefore, not commonly conducted in real-world clinical practice. However, information on cytogenetic risk was used to the best possible extent in the R-ISS variable. Third, there are additional eligibility criteria applied to patients in CARTITUDE-1, which could not be applied to select the cohort from Therapie Monitor (e.g., absence of certain cardiac conditions, absence of prior history of central nervous system involvement or signs of meningeal involvement of multiple myeloma), as these variables were outside of the scope of Therapie Monitor. Therefore, some patients within the RWCP group may have certain comorbidities that could not be adjusted for in this analysis.

## 5. Conclusions

In summary, findings from the analyses presented in this study demonstrate improved TTNT and OS for heavily pretreated or refractory patients with triple-class exposed MM compared with patients treated with current clinical practice based on real-world control data from Germany. These results highlight cilta-cel’s potential as a novel and effective treatment option to address the unmet treatment needs in triple-class exposed RRMM patients.

## Figures and Tables

**Figure 1 cancers-13-05996-f001:**
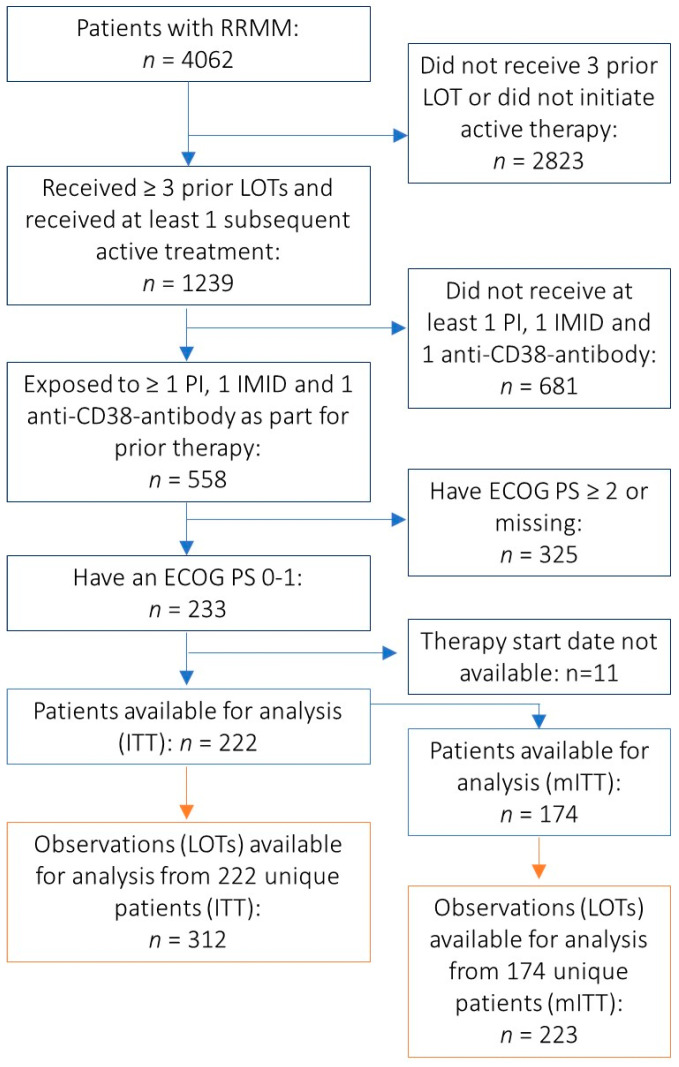
Patient Selection for Therapie Monitor Cohort from Therapie Monitor. Abbreviations: ECOG PS, Eastern Cooperative Oncology Group performance status; IMID, immunomodulatory drug; ITT, intention to treat; LOT, line of therapy; PI, proteasome inhibitor; RRMM, relapsed and refractory multiple myeloma.

**Figure 2 cancers-13-05996-f002:**
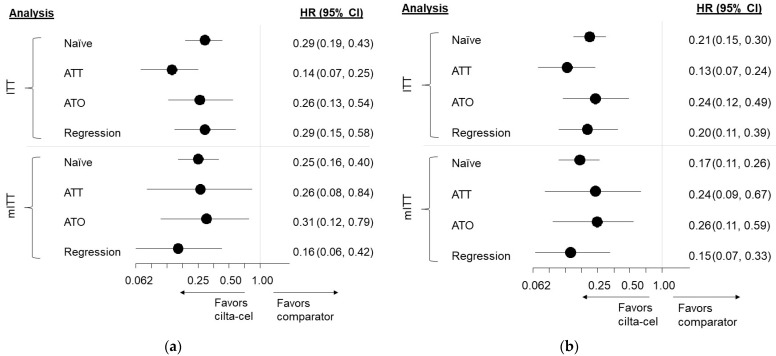
Findings from Adjusted Comparisons, OS and TTNT. Forest plots presenting hazard ratios comparing cilta-cel and RWCP with corresponding 95% CIs for (**a**) OS and (**b**) TTNT are shown for both ITT and mITT populations. Both the unadjusted comparison as well as adjusted comparisons derived using IPW methods and Cox proportional hazards regression are shown. Findings for both endpoints consistently suggest increased benefits with cilta-cel compared to RWCP. Abbreviations: ATT = average treatment effect in the treated population; ATO = average treatment effect in the overlap population; CI = confidence interval; HR = hazard ratio; ITT = intention to treat; mITT = modified intention to treat.

**Figure 3 cancers-13-05996-f003:**
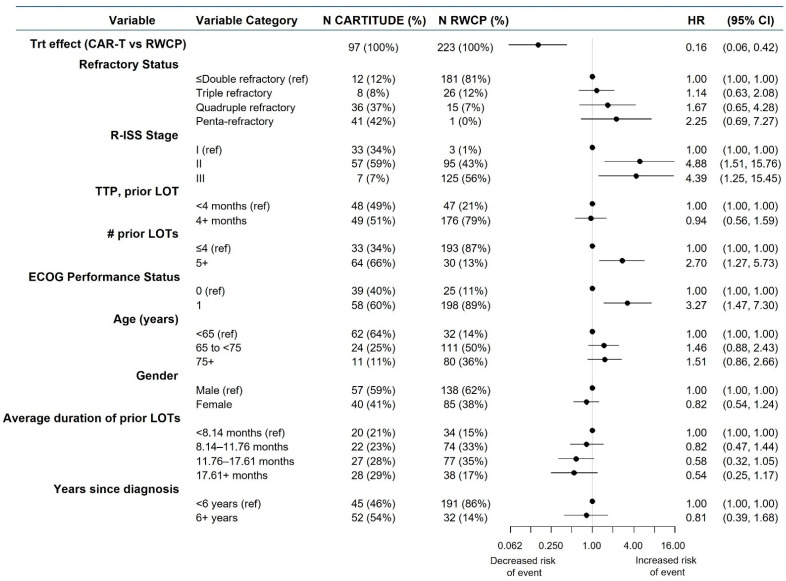
Overall Survival, mITT comparison, Covariate Effects from Cox PH Multivariable Regression Analysis. The clinical effects associated with the modeled covariates in a multivariable Cox proportional hazards regression model for OS are presented based upon data from the mITT population. R-ISS stage was derived for both cohorts based on individual components. Abbreviations: CI, confidence interval; LOT, line of therapy; HR, hazard ratio; R-ISS, Revised International Staging System; RWCP, real-world clinical practice; Trt, Treatment.

**Figure 4 cancers-13-05996-f004:**
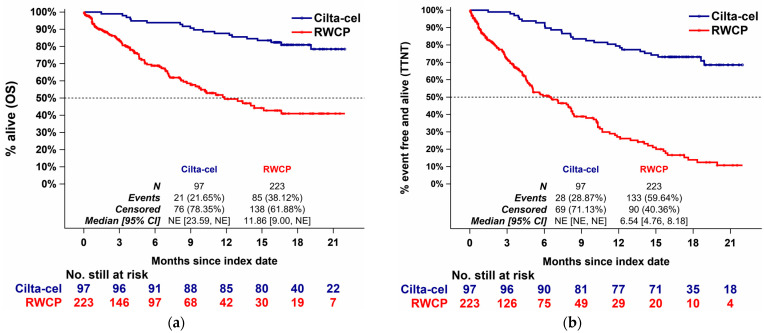
Unadjusted Kaplan–Meier Curves, OS and TTNT. Unweighted Kaplan–Meier curves in the cilta-cel and RWCP groups for the mITT population are shown for (**a**) OS and (**b**) TTNT. As these graphs present the unadjusted Kaplan–Meier curves, any differences in baseline characteristics as described in Figure 3 are not accounted for. For results of the adjusted comparisons, Figure 2 should be considered. Abbreviations: CI, confidence interval; OS, overall survival; NE, not estimable (not reached); RWCP, real world clinical practice; TTNT, time to next treatment.

## Data Availability

Data used for this study were based on CARTITUDE-1 and Therapie Monitor. CARTITUDE-1 data sharing is governed by the Janssen Pharmaceutical Companies of Johnson & Johnson data sharing policy that is available online. As noted on policy, requests for access to the study data can be submitted through Yale Open Data.

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
