# Peer review of "Adjusted Comparison of Outcomes between Patients from CARTITUDE-1 versus Multiple Myeloma Patients with Prior Exposure to PI, Imid and Anti-CD-38 from a German Registry"

_cancers, 2021, doi:10.3390/cancers13235996_

Round 1
Reviewer 1 Report
This is a statistically appropriate set of comparisons of outcome of patients enrolled in the Cartitude-1 trial with that of patients enrolled in an observatory multicenter registry. Nevertheless the definition of the reference population and the choice of variables to be introduced in the propensity score should be adequately reported or discussed :
Were exclusion criteria of the cartitude1 trial taken into account for selecting patients of the TherapieMonitor registry ? Cartitude-1 patients had to be free of any therapy targeted to B-cell maturation antigen (BCMA). Recent Allogeneic transplantation, active GVHD as well as recent autologous transplantation were exclusion criteria. More importantly, cardiac comorbidity was an exclusion criteria.
CNS and/or meningeal involvment as well as the need of immediate corticosteroid therapy were also exclusion criteria that should also be taken into account for the identification of the TherapieMonitor population. Patients with CNS and/or meningeal involvment or requiring immediate corticosteroid therapy may be at higher risk of early event.
Before weighting there was marked difference between the Cartitude-1 series and theTherapie Monitor population especially regarding one of the most important prognostic factor : R-ISS. Therefore showing the unadjusted Kaplan Meier curves in figure 4 seems inappropriate. These curves may be removed.
Did the propensity score logistic regression took into account potential differences in the comorbidities of patients as frequently observed between real-life population reports and clinical trial ? If not this point should be addressed in the discussion.
Page 4 line 174 : what is total plasmocytomas ? total number of bone lesion ? Was the extent of bone marrow infiltration available in the TherapieMonitor population?
Minor :
It is not always easy to read the manuscript because of the high number of abbreviations, sometimes rarely repeated (IPD), sometimes not introduced (LOT, EHR, SMD in appendix B ..)
Some references should be more accurately reported for example 33, 35
Author Response
Reviewer 1: “This is a statistically appropriate set of comparisons of outcome of patients enrolled in the Cartitude-1 trial with that of patients enrolled in an observatory multicenter registry.”
Authors: Thank you very much for your review and your positive feedback.
Reviewer 1: “Nevertheless the definition of the reference population and the choice of variables to be introduced in the propensity score should be adequately reported or discussed:
Were exclusion criteria of the cartitude1 trial taken into account for selecting patients of the TherapieMonitor registry ? Cartitude-1 patients had to be free of any therapy targeted to B-cell maturation antigen (BCMA). Recent Allogeneic transplantation, active GVHD as well as recent autologous transplantation were exclusion criteria. More importantly, cardiac comorbidity was an exclusion criteria.
CNS and/or meningeal involvment as well as the need of immediate corticosteroid therapy were also exclusion criteria that should also be taken into account for the identification of the TherapieMonitor population. Patients with CNS and/or meningeal involvment or requiring immediate corticosteroid therapy may be at higher risk of early event.”
Authors: A reduced set of the CARTITUDE-1 inclusion criteria were used to select the patients in the TherapieMonitor cohort. Specifically, these were (cf. lines 127-135, Track changes document)
(1) had been exposed to a PI, an IMiD, and an anti-CD38 MoAB as part of previous therapy (either from different monotherapies or combination regimens);
(2) had received at least 3 prior lines of MM treatment regimens (RRMM as defined by IMWG consensus criteria);
(3) received a subsequent therapy after becoming triple class exposed
and (4) had an ECOG score <2
However, as Reviewer 1 correctly points out, not all CARTITUDE-1 eligibility criteria were used to create the comparator cohort. TherapieMonitor is a real-world database, and it is in the nature of real-world data that they do not correspond to the requirements for the clinical testing and assessment with respect to the granularity and frequency as in in clinical trials. In TherapieMonitor all relevant real-world variables of a patient’s treatment journey surrounding multiple myeloma are collected. Therefore, the data focused on cardiac, systemic and nervous pathophysiology were not in scope and could not be used as inclusion criteria to create the comparator cohort. This is a limitation of the current comparison and we have therefore further clarified this in both the methods section (lines 135-139, Track changes document) and considered in the discussion (lines 396-402, Track changes document).
Reviewer 1: “Before weighting there was marked difference between the Cartitude-1 series and theTherapie Monitor population especially regarding one of the most important prognostic factor : R-ISS. Therefore showing the unadjusted Kaplan Meier curves in figure 4 seems inappropriate. These curves may be removed.”
Authors: We believe that these curves contain valuable information for the reader, especially on the shape of the two curves and the number of patients at risk at the different timepoints. We further believe that these figures are labeled clearly as unadjusted and are to be considered in context with the forest plots shown in Figure 2, where both the results for the unadjusted and adjusted comparison are shown. This allows the reader to contextualize the hazard-ratios of the adjusted comparisons.
Given we see value in presenting the graphs and this was not a concern for the other reviewers, we would prefer to keep these graphs. However, we do understand Reviewer 1’s point of view, and we can remove these plots if this is also preferred by the Editors.
Reviewer 1: “Did the propensity score logistic regression took into account potential differences in the comorbidities of patients as frequently observed between real-life population reports and clinical trial ? If not this point should be addressed in the discussion.”
Authors: The models shown in the manuscript took into account the following variables: refractory status, revised ISS stage (R-ISS) at index date, time to progression on last prior line, number of prior lines of treatment, ECOG status, age at index date, sex, average duration of prior lines and years since diagnosis.
While comorbidities are collected on a more general level in TherapieMonitor, it is in the nature of real-world data existing in the patient files that they do not correspond to the requirements of clinical testing and assessment with respect to the granularity and frequency as in clinical trials. Therefore, comorbidities were not used as a factor in the model.
This is a limitation of the comparison and we have hence made this explicit in both the methods section (lines 184-188, Track changes document) and considered in the discussion (lines 390-396, Track changes document).
Reviewer 1: “Page 4 line 174 : what is total plasmocytomas ? total number of bone lesion ? Was the extent of bone marrow infiltration available in the TherapieMonitor population?”
Authors: As total plasmacytomas all extramedullary plasmacytomas plus any soft-tissue part of bone-based plasmacytoma are considered. Extramedullary plasmacytomas were not recorded in TherapieMonitor. “Bone involvement” was a variable available in TherapieMonitor. Per CARTITUDE-1 protocol, bone lesions were required to be confirmed either by skeletal radiography, CT, or PET-CT. As outlined above, due to the real-world nature of the comparator cohort similarity of this variable was uncertain and therefore not included. As for the other limitations of this comparison, these are outlined in the manuscript’s method second (lines 184-188, Track changes document) and discussion (lines 388-396, Track changes document).
Reviewer 1: “It is not always easy to read the manuscript because of the high number of abbreviations, sometimes rarely repeated (IPD), sometimes not introduced (LOT, EHR, SMD in appendix B ..)”
Authors: Thank you very much for your comment. We have added further definitions where appropriate and removed any abbreviations if these were used less than three times in the manuscript.
Reviewer 1: “Some references should be more accurately reported for example 33, 35”
Authors: References in the revised manuscript are presented as per the journal’s style guide. Further, the authors consider text referenced by 33 and 35 as appropriate. We would therefore welcome further clarity in how we can include these more accurately.
Reviewer 2 Report
In this manuscript, authors successfully compared and evaluated the effectiveness of cilta-cel for overall survival and time to next treatment with real-world clinical practice.
Except the resolution of tables and font size (needs to be bigger for easier read) there is nothing I can suggest to change in this manuscript.
Author Response
Reviewer 2: “In this manuscript, authors successfully compared and evaluated the effectiveness of cilta-cel for overall survival and time to next treatment with real-world clinical practice.”
Authors: Thank you very much for your review and your positive feedback.
Reviewer 2: “Except the resolution of tables and font size (needs to be bigger for easier read) there is nothing I can suggest to change in this manuscript.”
Authors: Thank you for your feedback. We have increased the font size for the different figures and used higher resolution images.
Reviewer 3 Report
In this work, Dr. Mertz and colleagues compare the outcomes of patients treated with standard-of-care options or with cilta-cel on the CARTITUDE-1 clinical trial.
The study is interesting since in the CARTITUDE-1 trial there was no comparison arm.
However, I will add some clarifications and improve the discussion. First, it is unclear to the reviewer what year of diagnosis was included in the “real-world” population arm. Are these patients diagnosed between 2016 and 2020 or prior?
Also the age in the two cohorts is unbalanced. Indeed, in the real-world population arm 76 percent of patients are older than 65, while in the CARTITUDE arm only 36 percent of the patients are older than 65. The real-world population cohort is more representative of the average MM patient but this should be addressed in the discussion. Older patients might be frailer, tolerate less therapies, or have poor outcomes for unrelated conditions.
Finally, cytogenetics risk is not included in the analysis but only as part of the R-ISS. Also in this case the real-world population includes patients with high-risk disease.
I will add cytogenetics risk analysis as a separate variable and discuss the differences in terms of age and stage in the discussion.
Author Response
Reviewer 3: “In this work, Dr. Mertz and colleagues compare the outcomes of patients treated with standard-of-care options or with cilta-cel on the CARTITUDE-1 clinical trial.
The study is interesting since in the CARTITUDE-1 trial there was no comparison arm.”
Authors: Thank you very much for your review and your positive feedback.
Reviewer 3: “However, I will add some clarifications and improve the discussion. First, it is unclear to the reviewer what year of diagnosis was included in the “real-world” population arm. Are these patients diagnosed between 2016 and 2020 or prior?”
Authors: Thank you for your comment. We have updated the text to reflect the follow-up more accurately (lines 111-116 in track changes document). Earliest date of diagnosis for patients included in TherapieMonitor is 22 August 2002, while it is 30 June 2001 for patients in CARTITUDE-1. In TherapieMonitor, longitudinal follow-up started in 2016, either with new diagnosis or inclusion of the patient in the registry. Information from before 2016 is retrospectively included in the database. As daratumumab only became available in 2016, all index dates (i.e., patients fulfilling the inclusion criteria of this comparison including being tri-exposed) for patients included in this study is after 2016. Below you find a histogram showing the date of diagnosis and index date for inclusion of the line of therapy in this study for CARTITUDE-1 and TherapieMonitor.
Accordingly, we edited the sentence in the manuscript removing that patients were followed since diagnosis, and highlighting that they were followed since 2016.
Reviewer 3: “Also, the age in the two cohorts is unbalanced. Indeed, in the real-world population arm 76 percent of patients are older than 65, while in the CARTITUDE arm only 36 percent of the patients are older than 65. The real-world population cohort is more representative of the average MM patient, but this should be addressed in the discussion. Older patients might be frailer, tolerate less therapies, or have poor outcomes for unrelated conditions.”
Authors: We agree with Reviewer 3 that there are significant differences between the two cohorts before applying the weighting approaches. However, as shown in Table S1, these differences are markedly reduced after applying ATT weights and are removed with ATO weights. Remaining imbalances after ATT weights are discussed in lines 388-390 (Track changes document). However, we have added further context to the differences and limitation of study in lines 390-402 (Track changes document).
Reviewer 3: “Finally, cytogenetics risk is not included in the analysis but only as part of the R-ISS. Also in this case, the real-world population includes patients with high-risk disease.
I will add cytogenetics risk analysis as a separate variable and discuss the differences in terms of age and stage in the discussion.”
Authors: Cytogenetic risk testing typically requires a bone marrow aspiration and is a costly procedure which is not commonly done in real-world clinical practice. Additionally, results from cytogenetic risk testing only have a limited influence on therapy choice in multiple myeloma. Missingness in cytogenetic risk is also high in TherapieMonitor, therefore, reflecting it as an additional factor in the R-ISS stage calculation is the best way cytogenetic risk can be reflected in the analysis, and is the rationale as to why it was not included as a separate variable in the model. We have further clarified the low proportion of observations for which a cytogenetic test was available in TherapieMonitor in lines 393-396 (Track changes document).
